# Diisocyanates influence models of atopic dermatitis through direct activation of TRPA1

**Manoj Yadav**[1], **Prem Prashant Chaudhary**[1], **Brandon N. D'Souza**[1], **Grace Ratley**[1], **Jacquelyn Spathies**[1], **Sundar Ganesan**[2], **Jordan Zeldin**[1], **Ian A. Myles**[1]*

**1** Epithelial Therapeutics Unit, National Institute of Allergy and Infectious Disease, National Institutes of Health, Bethesda, Maryland, United States of America, **2** Biological Imaging Section, Research Technology Branch, NIAID, NIH, Bethesda, Maryland, United States of America

* mylesi@niaid.nih.gov

**Data Availability Statement:** All relevant data are within the paper and its Supporting information files.

## Abstract

We recently used EPA databases to identify that isocyanates, most notably toluene diisocyanate (TDI), were the pollutant class with the strongest spatiotemporal and epidemiologic association with atopic dermatitis (AD). Our findings demonstrated that isocyanates like TDI disrupted lipid homeostasis and modeled benefit in commensal bacteria like *Roseomonas mucosa* through disrupting nitrogen fixation. However, TDI has also been established to activate transient receptor potential ankyrin 1 (TRPA1) in mice and thus could directly contribute to AD through induction of itch, rash, and psychological stress. Using cell culture and mouse models, we now demonstrate that TDI induced skin inflammation in mice as well as calcium influx in human neurons; each of these findings were dependent on TRPA1. Furthermore, TRPA1 blockade synergized with *R. mucosa* treatment in mice to improve TDI-independent models of AD. Finally, we show that the cellular effects of TRPA1 are related to shifting the balance of the tyrosine metabolites epinephrine and dopamine. This work provides added insight into the potential role, and therapeutic potential, or TRPA1 in the pathogenesis of AD.

## Introduction

Atopic dermatitis (AD) is a chronic inflammatory skin disease most often enumerated on the severity of the presenting rash [1]. However, itch is by far the more disruptive symptom to patients and caregivers, in part, through its contribution to the marked psychosocial burdens of AD [2]. Recent work from several groups has elucidated the neuro-endocrine mechanisms of the itch associated with AD [3–5]. These discoveries include a new appreciation for the role of the transient receptor potential (TRP) family ion channels. Canonic activations of the TRP receptors occurs with temperature changes but they can also be modulated by natural compounds. For example, a recent Nobel prize was awarded to the discovery that capsaicin from peppers generates the sensation of heat through activations of TRP Vanilloid 1 (TRPV1) [6]. Another cutaneous channel in this class, TRP Ankyrin 1 (TRPA1) is typically activated by temperatures below 17°C, but also modulates itch, inflammation, as well as murine models of anxiety and depression [7, 8].

**Funding:** This work was supported by the Intramural Research Program of the National Institute of Allergy and Infectious Diseases (NIAID). The funders had no role in study design, data collection and analysis, decision to publish, or preparation of the manuscript.

**Competing interests:** The authors have declared that no competing interests exist.

We recently reported that an untargeted comparison of Environmental Protection Agency (EPA) pollution databases versus AD rates by US zip codes revealed that isocyanates, such as toluene diisocyanate (TDI), is the pollutant class with the strongest spatiotemporal association with AD [9]. We further demonstrated exposure of health-associated commensal bacteria to TDI shifts the microbial metabolism towards an AD-like phenotype; both TDI and hydrogen isocyanate disrupted lipid homeostasis in and negated the modeled benefit of commensal isolates of *Roseomonas mucosa* through disrupting nitrogen fixation [9]. However, beyond the impacts on commensal bacteria, the discovery that TRPA1 is directly activated by TDI [10–15] suggested that this pollutant may also contribute to AD through directly impact the host.

Herein we demonstrate that, like asthma models [10, 12], TDI-induced dermatitis models are partially TRPA1 dependent. Blockade of TRPA1 synergized with *R. mucosa* to improve outcomes in both TDI- and MC903-mediated AD mouse models. Using untargeted metabolomics, we identified that the effects of TRPA1 were most strongly associated with changes in tyrosine-related metabolic balance between epinephrine and dopamine. Overall, these data suggest TRPA1 blockade may offer an additional therapeutic target in AD-associated itch.

## Materials and methods

### Mice

Male and female B6129PF2/J as well as TRPA1 knock out mice aged 6–12 weeks were purchased from Jackson Labs (Bar Harbor, MA). All mice were age and sex matched within each experiment. The MC903 model was performed as previously described [16]. We verified that the effects of interventions were present in both male and female mice but did not compare between sex groups directly. For TDI exposure, TDI (Sigma Aldrich, St Louis MO) was diluted in acetonitrile (Sigma) to a 2% solution; 10 microliters were applied topically to each ear of the mice every other day for three weeks. The ears were collected and analyzed as described for the MC903 model. For experiments using imaging mass spectrometry, mouse tissue was fresh frozen and analyzed on positive ion mode as previously described [17]. By ARRIVE guidelines [18], study design was a comparison between either wild type (WT) or TRPA1-/- mice challenged with TDI, or WT mice in the MC903 model treated with combinations of topical *R. mucosa*, TRPA1 blockers, or diluent controls. No randomization, inclusion/exclusion criteria, or blinding were used. N values were calculated based on previous results for *R. mucosa* treatment in mice [16]. Assuming an 8% mean improvement in swelling in the diluent group and a 20% ± 5% in the treatment group would require 4 mice per group based on two independent groups measured by continuous differences. Thus, each experiment used a minimum of 4 per experiment but were combined to assess consistency between experiments. Outcomes measured were ear thickness, which is a sign of swelling which indicates worse outcomes in the models. Statistical comparisons were either Student t test for two-group experiments, or ANOVA for multi-group experiments. Animal work was approved by an Institutional Animal Care and Use Committee (IACUC) and followed the guidelines and basic principles in the United States Public Health Service Policy on Humane Care and Use of Laboratory Animals, and the Guide for the Care and Use of Laboratory Animals by certified staff in an Association for Assessment and Accreditation of Laboratory Animal Care (AAALAC) International accredited facility. Animals were anesthetized using isoflurane and sacrificed using $CO_2$ and cervical dislocation per AAALAC guidelines. All animals were monitored for signs of suffering and, if any had been present, mice would have euthanized to avoid prolonged suffering.

## Cell culture scratch assay

HaCaT keratinocytes cultured in KFSM (Gibco, Billings MT) media were seeded at a density of 50,000 cells/well in 8 well chamber labtek culture slides (Lab-Tek II, CC2. ThermoFisher Scientific). Schwann cells were cultures in Schwann cell medium (#1701) (ScienCell research Laboratories) and seeded in an identical manner. After 24 hr of incubation at 37˚C and 5% $CO_2$ a pipette tip was used to scratch the cells with two to six experiment groups and one control per trial. After the scratch was created, the media was changed, and cells were incubated for an additional 24hr allowing cell migration and proliferation to close the scratch. Monitoring of the wound closure was performed using the Cytation 5 and their Gen5 software per manufacturer instructions (BioTek, Winooski, VT).

## Calcium influx assessment

Schwann cells were seeded in the glass bottom dishes for the $Ca^{2+}$ imaging experiment. The next day cells were loaded with a calcium sensitive dye (Fluo-4 AM, 2 μM for 30 min; Thermo-Fisher). After cells were washed with warm PBS and added the warm fresh culture media. The cells were imaged with Leica SP8 confocal microscope. Cells were stimulated with specific agonists or antagonists (TDI used at 7mM, 0.7mM, or 70nM) or diluent control as described during the imaging. Blockade of TRPA1 with HC030031 (Sigma Aldrich Inc) was performed at 10mM 30min before and/or immediately prior to challenge with TDI. The live images were quantified using ImageJ software and graphs were generated with GraphPad Prism software.

## Bacterial culture

Isolates of *R. mucosa* were selected as previously described [16, 19, 20]. *Roseomonas* was grown in R2A broth (Teknova; Hollister, CA) or on R2A agar (Remel; San Diego, CA) at 32˚C. For topical application on mice ears *R. mucosa* was diluted in the 10% sucrose solution and prepared the working solution at OD 0.4. *R. mucosa* solution and diluent 10% sucrose solution was applied on both ears as described.

## Matric assisted laser desorption ionization (MALDI) tissue imaging

Fresh frozen tissue collection, matrix spraying, data acquisition and MALDI imaging were performed as described previously [17]. The matrix solution used was 20 mg/mL 5-dihydroxybenzoic acid (#149357-20G Sigma-Aldrich Inc, Missouri, USA) in 100% acetone and 0.1% trifluoroacetic acid (#302031 Sigma). The tissue was scanned with both MS and TIMS settings at a resolution of 20 μm. The MS settings were: scan range 20–2500 m/z in positive MS scan mode. The TIMS settings were: 1/K0 0–8 − 1.89 V×s/cm2, ramp time of 200 ms, acquisition time of 20 ms, duty cycle = −10%, and ramp rate of 4.85 Hz. Acquired raw data were initially processed with SCiLS lab 2021a (Bruker Scientific LLC, Billerica, MA, USA) and the file was exported to Metaboscape 2021b (Bruker, USA) for annotations and further downstream analysis.

## Non-metric multidimensional scaling (NMDS) plots and heatmaps

MALDI imaging data was exported from Metaboscape 2021b for further statistical analysis in R. Nonmetric dimensional scaling was carried out in R by using the vegan package. To calculate the statistical difference between the concentration of a metabolite in the groups tested, ANOSIM was employed. For heatmap generation, p-values (FDR) were calculated by t-tests between the two groups, and we plotted a heatmap of the top 100 most significantly different metabolites between the two groups. Heatmap library was used to plot the heatmap. After the

peak intensity table was imported, the uploaded data were log-transformed, and normalization was performed by mean subtraction. Other parameters that were set included the use of correlation-based clustering of the columns. To simplify the visualization of the abundances of metabolites across the treatments, the top 100 metabolites ranked by t-test were shown.

## Metabolic pathway analysis

Pathway identification was performed using metaboAnalyst (https://www.metaboanalyst.ca/) which determines the probability of a pathway being precent based on MS1 features associated with multiple metabolites within the reaction hierarchy of a specific pathway. The output of MetaboAnalyst (functional analysis), index of pathway significance (IPS) values was calculated using the formula: $((\text{Significant hits} + 1)^2 + (\text{Total} - \text{Significant})) / (\text{FET}^2 * \text{Pathway Metabolites} * \text{Expected Hits})$. This formula consolidates five metrics provided by MetaboAnalyst that, when considered together, quantifies the pathways differences between two samples. A higher IPS value signifies a greater difference between the samples with respect to that pathway, and the use of a single values allows for efficient comparisons between pathways or between samples. The numerator consists of the weighted sum of significant and non-significant metabolites that were found in the samples of interest and divides it by the total number of metabolites in that same pathway. To avoid calculating the same IPS value when the number of significant metabolites is zero and one, respectively, one is added to the number of identified metabolites deemed to be significant. This ratio is then divided by the number of metabolites from that pathway that MetaboAnalyst expected to find in the samples, with a lower value further underscoring the significance of any and all identified metabolites. With FET or p-value being perhaps the metric most indicative of differences between two samples, it is squared and positioned in the denominator to increase the final IPS score exponentially as it approaches or passes statistical significance. By using these IPS values Pathway heatmaps were generated by using pheatmap package in R version 4.1.0. statistical significance was determined based on log10 IPS values.

## Statistics

The Student t test was used for two group comparisons whereas multi-group comparisons were performed using ANOVA. The respective figure legends indicate the test used. All statistics were performed using PRISM (GraphPad) as indicated in the respective figure legends.

## Study approval

All mouse experiments were approved and monitored by the animal control protocols at the National Institutes of Health.

## Results

### TDI-induced dermatitis is partially TRPA1 dependent

Prior work has established mouse models of TDI-induced asthma are TRPA1 dependent [10, 12]. However, despite TDI also being an established inducer to AD-models [21], the role of TRPA1 was unclear. We found that TRPA1[-/-] mice challenged with TDI (Fig 1A) had reduced swelling and reduced vascularization compared to wild type (WT) mice (Fig 1B and 1C); each of these are signs of improved outcomes in the model. Histologic examination confirmed reduced inflammatory infiltrate and epidermal thickening in TRPA1[-/-] mice (Fig 1D). There were no baseline differences in the WT versus TRPA1[-/-] mice (S1 Fig), indicating that the TDI-induced dermatitis was TRPA1 dependent. In addition, consistent with direct effects on

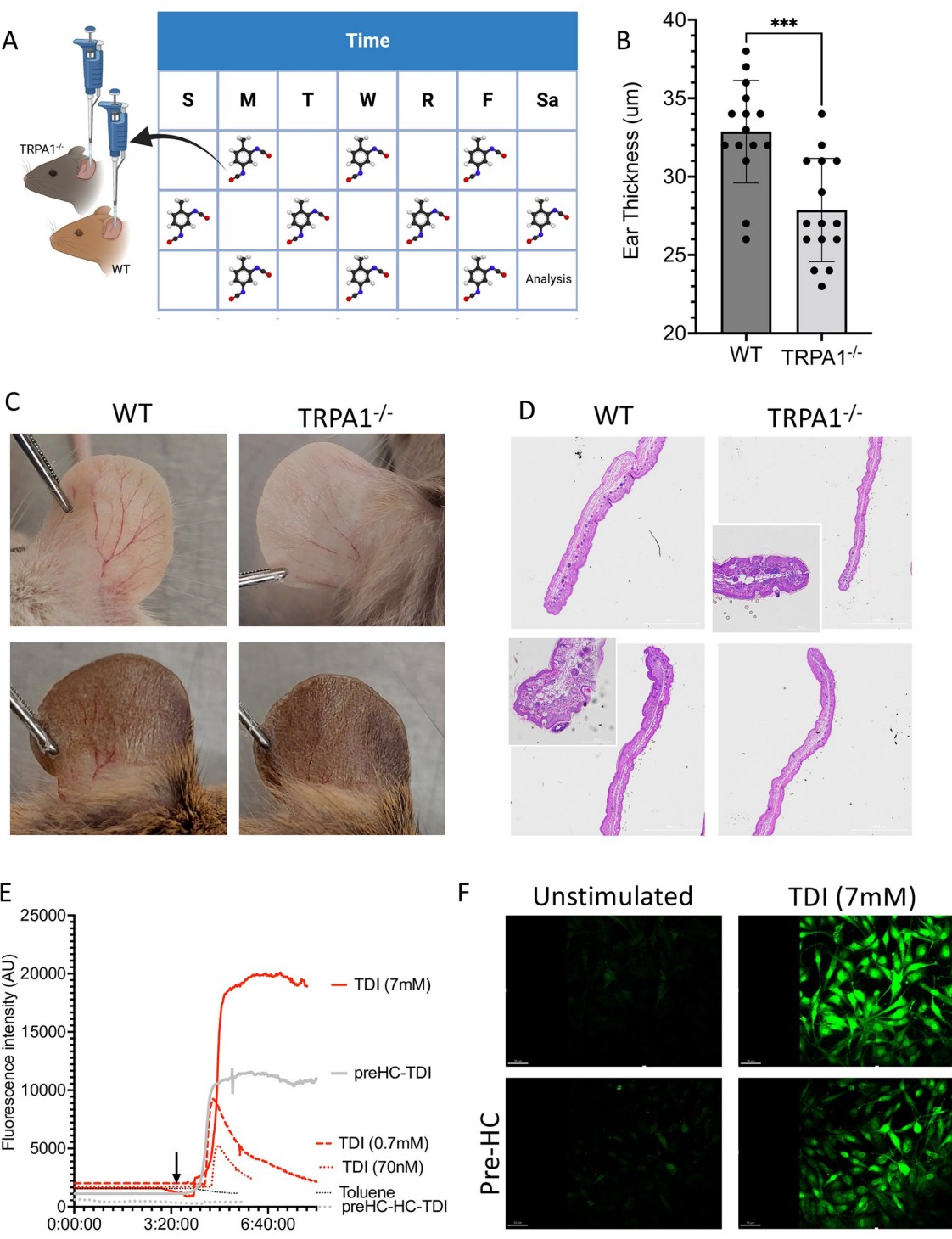

**Fig 1. Toluene diisocyanate (TDI) induced dermatitis in mice is dependent on TRPA1.** (A) Overview of model where TDI was applied to B6129PF2/J wild type (WT) or TRPA1 knockout (TRPA1-/-) mouse ears every other day for 3 weeks. (B-D) Resultant thickness for each mouse (B), representative photos (C) and H&E histology (D) are shown. Note that both the B6129PF2/J control and TRPA1-/- mice have natural variation in coloration, thus a representative image from various mice are shown to encompass the range in these strains. (E-F) Schwann cells stimulated with either TDI (at indicated final concentrations) or toluene (arrow indicates when TDI was added). In select conditions pre-incubation with the TRPA1 inhibitor HC030031 either 30 minutes prior (preHC), immediately prior (HC), or both (preHC-HC). Mean calcium flux fluorescent intensity (E) and representative image at peak brightness (F) are shown. Full data set and images can be found in S1 Fig and S1 Video. Data represent three independent experiments and are displayed as mean+SEM (B) or mean (E). *** indicated p < 0.001 as determined by Student T test.

TRPA1, the dose-dependent TDI induced induction of $Ca^{2+}$ influx in Schwann cells was inhibited by the TRPA1 blocker HC030031 (Fig 1E and 1F; S1A–S1C Video).

## TRPA1 blockade improved TDI-independent models of atopic dermatitis

To assess if TRPA1 may influence skin repair beyond the direct effects of TDI, we also tested TRPA1 modulation in TDI-free murine models of AD. The MC903 model of AD using topically applied calcipotriol hydrate to induce AD-like dermatitis [16]. After the dermatitis was induced, we topically treated mice with either diluent or inhibitors of TRPA1 (cardamonin and its natural source, cardamom seeds) (Fig 2A). Treatment with cardamom seeds, but not purified cardamonin reduced swelling and inflammation in mice (Fig 2B and 2C). However,

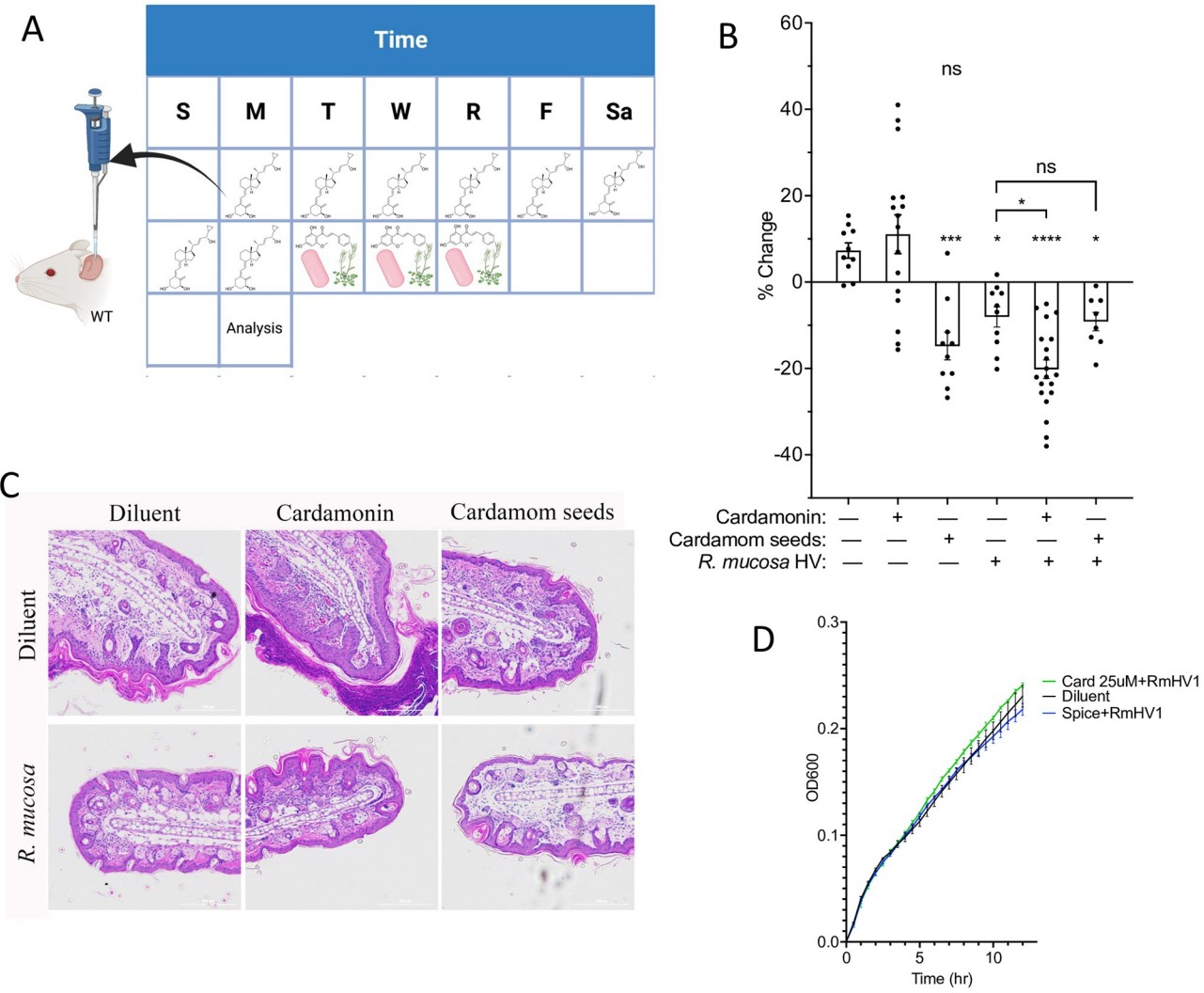

**Fig 2. TRPA1 blockade improves non-TDI models of atopic dermatitis.** (A) Overview of MC903 model where topical application occurred daily for 8 days before topical *Roseomonas mucosa*, cardamonin (Card), or ground cardamom seeds (Spice) were applied for three days. Ears were assessed 3 days after treatment; resultant ear thickness (B) and representative histology (C) are shown (N = 8–19 per group with each dot representing one mouse). (D) *R. mucosa* was grown in broth with either diluent, cardamonin or ground cardamom seeds (N = 1 isolate in triplicate wells). Resultant impact on optical density is shown. Data represent a combination of (B) or representative of (C-D) two or more independent experiments and are displayed as mean+SEM. ns = not significant, * = p <0.05, ** = p <0.01, *** = p <0.001, **** = p<0.0001 as determined by ANOVA for all displayed items.

cardamonin synergized with *R. mucosa* treatment (Fig 2B and 2C) despite no impact on *R. mucosa* growth (Fig 2D).

## TRPA1 modulated tyrosine metabolism

To assess potential mechanisms of TRPA1 on TDI-induced dermatitis, we assess the ear tissue by imaging mass spec on a matrix-assisted laser desorption ionization instrument (MALDI) (Fig 3A). Comparing the metabolomic signals from TDI-challenged mouse ears from WT mice and TRPA1$^{-/-}$ revealed general overall similarity (Fig 3B) but also showed several distinguishing metabolites identified by both m/z and collisional cross section (CCS) (Fig 3C). In TRPA1 blockade, similar findings of overall similarity (Fig 4A and 4B) with select differences (Fig 4C) were seen.

## TRPA1 effect on wound cultures are modulated by catecholamine signaling

Bundling the identified features from MALDI analysis into pathways using the Metaboanalyst software [22] revealed that tyrosine metabolism was the most upregulated pathway in TRPA1$^{-/-}$ mice (Fig 5A). Tyrosine metabolism was also the most upregulated pathway when comparing the adjunct effects of TRPA1 blockade versus *R. mucosa* (Fig 5B–5D). The shared impact of tyrosine suggested a possible role for modulation of catecholamines (dopamine, epinephrine, and norepinephrine), given that each is a derivative of tyrosine [23]. We used the *in vitro* scratch assay model, in which cells are grown to confluence, are physically "scratched", and then monitored to assess the time it takes to close the resultant cellular gap through a combination of cell proliferation and migration. We have previously shown that this model correlated with our therapeutic outcomes in both clinical trials of AD and the MC903 mouse model [16]. Activation with the TRPA1 agonist cinnamaldehyde [24] reduced modeled healing in both the HaCaT keratinocyte (KC) and Schwann (neuronal) cell lines (Fig 5E and 5F). Blockade of TRPA1 improved healing times in Schwann cells but not in KC (Fig 5E and 5F). In Schwann cells, the effect of cinnamaldehyde were partially reversed by the dopamine D2 receptor blocker haloperidol but was enhanced by co-administration of the anti-adrenergic sotalol (Fig 5G).

## Discussion

The interaction between TDI and TRPA1 presents an intriguing pathology suggesting a role in allergic disease. TDI is a TRPA1 agonist [15] and component of numerous exposures that are known to increase the incidence and severity of AD ranging from fabrics like nylon and polyester to home remodeling chemicals like paint, polyurethane, and memory foam furniture [9]. TDI is a TRPA1-dependent inducer of other atopic models such as contact dermatitis [3], asthma [10, 12], allergic rhinitis [25, 26], and chronic itch [27]. TDI and the related hexamethylene diisocyanate (HDI) are two of the fewer than ten chemicals used to induce AD in mouse models [21]. Although untested in skin cells, *in vitro* models using bronchial epithelial cells demonstrated that TDI erodes barrier function through effects on barrier proteins such as occludin [28]. In mice, TRPA1 and TRPV1 are upregulated in the skin after consumption of the AD-associated "Western diet" [29, 30], potentially suggesting the dietary and gut microbiome findings associated with AD [31–33] may sensitize patients to the deleterious effects of TDI.

TRPA1 co-activation is required for the activity of one of the central allergic mediators, thymic stromal lymphopoietin (TSLP) [34]. In cultures of neurons and mast cells, TRPA1 can create a positive feedback loop with the allergic cytokine IL-13 [35]. Furthermore, in mice, TRPA1 governs models of anxiety and depression [8] which are both strongly associated

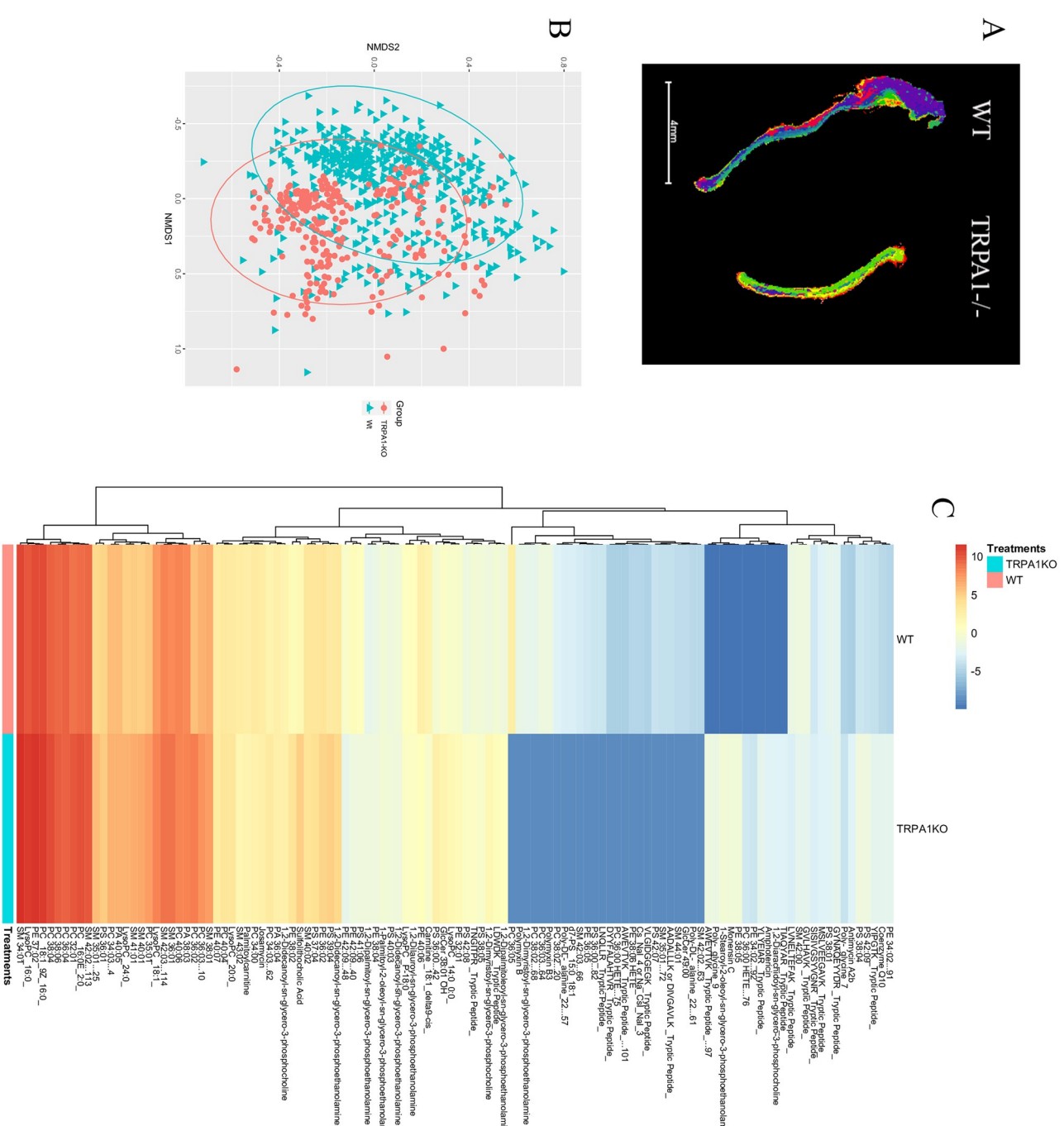

**Fig 3. TRPA1$^{-/-}$ have differing metabolic response to TDI exposure.** Ears from mice treated with TDI as in Fig 1 were collected and examined by imagine mass spec. (A) Segmentation for representative ear, (B) NMDS similarly plot, and all of the metabolites that were identified by both m/z and collisional cross section (C) are shown. Data represent three independent experiments and calculated as NMDS (B) or ANOVA (C). WT = wild type mice.

comorbidities of AD [36]. While uncontrolled itch induces secondary psychiatric stress on its own, these recent elucidations of TRP receptors present the possibility that the association between itch and psychological symptoms could partially reflect the dual consequences of TRPA1 activation.

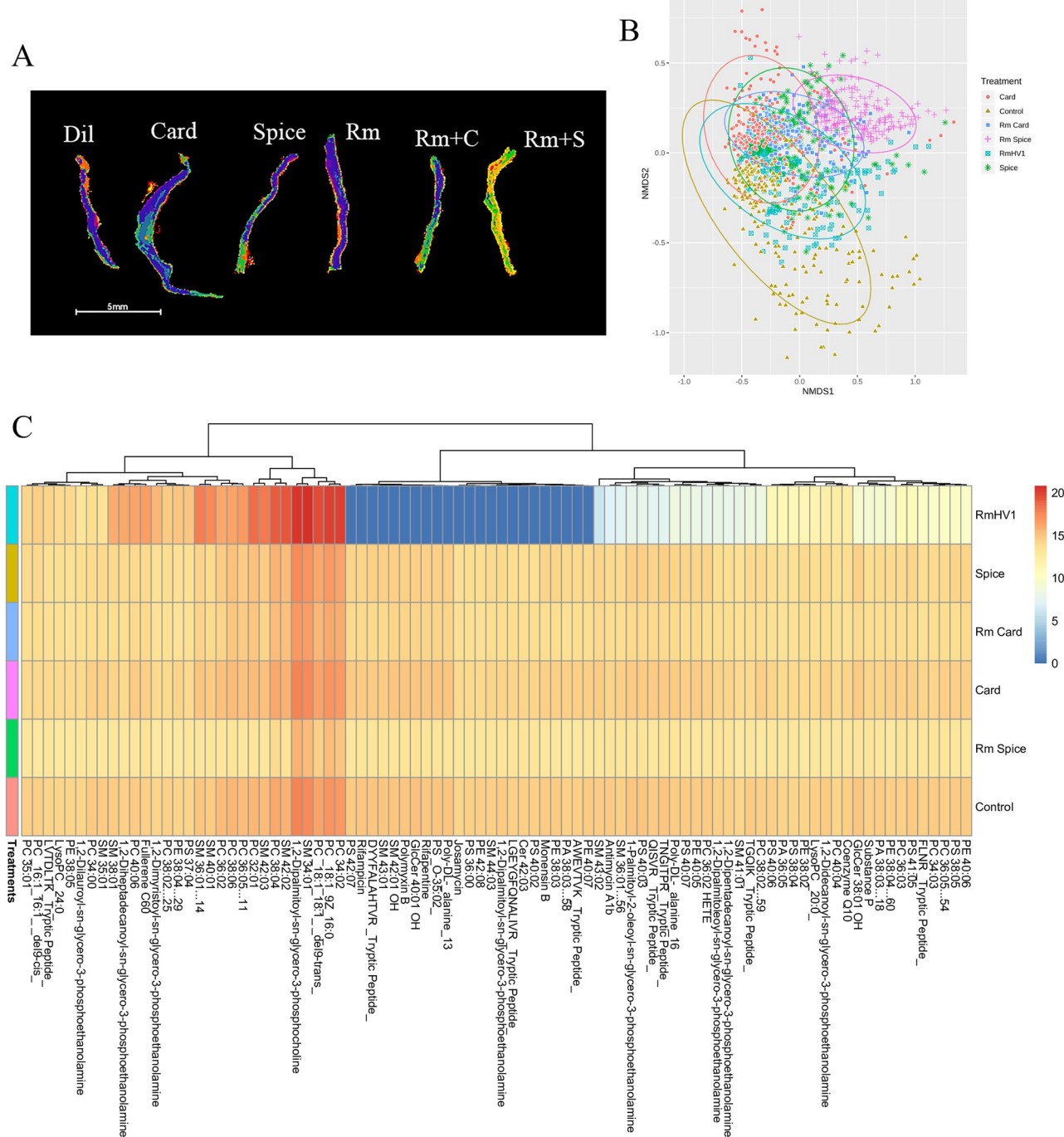

**Fig 4. TRPA1 blockade alters metabolic response to AD models in mice.** Ears from mice treated with TDI as in Fig 3 were collected and examined by imagine mass spec. (A) Segmentation for representative ear, (B) NMDS similarly plot, and all of the metabolites that were identified by both m/z and collisional cross section (C) are shown. Data represent two independent experiments and calculated using ANOVA. Dil = diluent, Card = cardamonin, Spice = ground cardamom seeds, Rm = Roseomonas mucosa, Rm+C = *R. mucosa* and Card, Rm + S = *R. mucosa* and Spice.

Our findings beget a working hypothesis in which (di)isocyanate exposure induces dysbiosis [9] through carbamoylation of the amine-groups on the ceramide-sphingolipid family of lipids [37] needed for proper skin barrier function [38]. While altering commensal metabolism, (di)isocyanates may also directly activate TRPA1 to induce itch, rash, Th2 cytokines, and

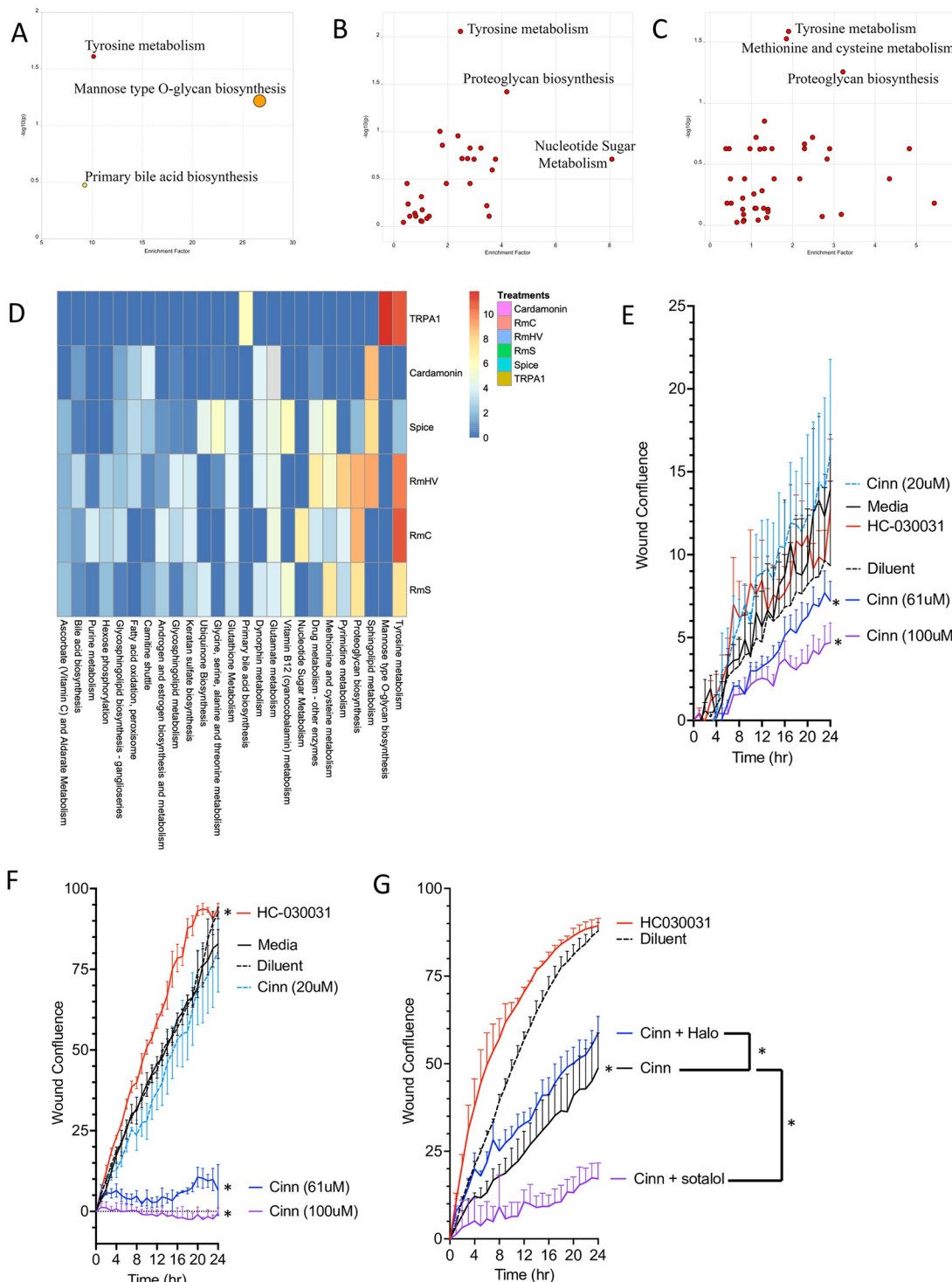

**Fig 5. Metabolic influence of TRPA1 is concentrated in tyrosine metabolism.** (A-C) Individual pathways as indicated from Metaboanalyst taken from metabolites identified as different between wild type and TRPA1-/- mice in response to TDI (A), *R. mucosa* treated versus *R. mucosa* plus Cardamonin (RmC; B), and *R. mucosa* treated versus *R. mucosa* plus ground cardamom seeds (RmS; C). (D) Summarizing all pathways impacted by index of pathway significance (IPS) for each condition versus its diluent control. (E, F) Percent of starting wound closure over time for keratinocytes (E) and Schwann neuron cells (F) incubated with diluent, the TRAP1 agonist cinnamaldehyde (Cinn) or TRPA1 blocker HC030031 (N = 1 cells line each in triplicate wells). Significance determined by comparison of area under the curve with 95% confidence intervals by PRISM. (G) Schwann cells in scratch assay with Cinn with or without addition of anti-dopaminergic haloperidol (halo) or anti-adrenergic sotalol (N = 1 cells

line each in triplicate wells). Data represent two or more independent experiments and are shown as mean ± SEM. * = p < 0.05 for comparison of area under the curve for wound closure as indicated.

further erode barrier function. A hypothesis which considered TRP receptors as an important activator of AD would be consistent with the description of AD as "the itch that rashes" [39] since peripheral itch from TRPA1 would send afferent signals of itch to the brain, but the resultant inflammatory mediators might not be released until the efferent signals were returned to the skin after the scratching occurred [15].

Our data is also consistent with prior reports linking dopaminergic responses with TRPA1 activity [40, 41]. Glucocorticoids, the most common AD treatment class, has been shown to upregulate dopamine beta-hydroxylase, which converts dopamine to epinephrine and norepinephrine [42]. Our working model suggests that dopamine and epinephrine may have opposing effects on TRPA1 activation and that effects of TRPA1 blockade may be partially reversed by dopamine receptor blockers while being enhanced by anti-adrenergic compounds. While this work furthers the reported link between TRPA1 and catecholamines, uncovering a full understanding of the mechanism linking TRPA1 activation, dopaminergic signaling, and eventual tissue repair will require additional studies.

The literature suggests that each of the chemical compounds which activate TRPA1 (TDI, mustard oil, thiosulfates, tear gas, and cinnamaldehyde) do so through direct alteration of cysteines [15, 43, 44]; cysteine is also a key amino acid regulated in nitrogenase during nitrogen fixation [45]. Therefore, a possible unifying mechanism may entail TDI activating TRPA1 while inactivating commensal nitrogen fixation through impacts on specific cysteine residues of TRPA1 and nitrogenase. However, further work will be needed to validate the mechanism of nitrogen fixation in *R. mucosa*.

While our study presents population-level associations along with a suggested mechanism, our work is limited in that we do not yet have direct patient-level evaluations. This work's major limitation is the lack of direct assessment of the differences in TRP expression by age and body site. Alterations in TRPA1 expression between patients with AD and controls, particularly at the young ages which are the critical window for AD risk, would help answer if pollutants create deleterious imprinting onto neuro- and/or immune-development through TRPA1. One study demonstrated a higher TRPA1 expression in the skin of patients with AD, however this was limited to only three adults [35].

Furthering our environmental correlations will also require the development of equipment capable of directly assessing air concentrations of these chemicals as well as directly assessing the surrounding incidence, prevalence, and severity of AD. With both longitudinal symptom and exposure data, the needed odds ratios for exposure and AD risk can be calculated on an individual level, rather than being limited to our current population-level associations.

Our pre-clinical evaluation of TRPA1 blockade is definitionally limited to mice and would require human clinical trials to validate. The promises and limitations of TRPA1 blockade have been well reviewed [46]. In brief, TRPA1$^{-/-}$ mice appear overall normal and, unlike the other TRP receptors, TRPA1 blockade does not harm systemic thermoregulation. TRPA1 blockade does however carry a theoretical risk of loss of the nociception, which could block the needed signals that would otherwise cause people to avoid certain irritants. However overall, this work suggests (di)isocyanates should be further assessed as a potential contributor to AD and presents TRPA1 a potential therapeutic target for this patient population.

## Supporting information

**S1 Video. TDI directly activates calcium flux from neurons in a TRPA1 dependent manner.** (A) Fluorescence intensity for calcium flux cause by toluene diisocyanate (TDI) added to Schwann neurons at time indicated by arrow. (B) As in A but cells were pre-treated with TRPA1 inhibitor HC030031 (HC) 30 minutes prior to TDI exposure. (C) Similar to A and B, cells were pre-treated with HC 30min prior and then re-exposed to HC immediately prior to TDI challenge. Data represent four independent experiments.
(MOV)

**S1 Fig. No baseline differences were present between wild type (WT) and TRPA1$^{-/-}$ mice.** (A) Left ear thickness from 8 mice aged 8 weeks for either WT or TRPA1-/- mice. (B) Representative images from two mice per group showing gross ear anatomy. (C) Representative H&E slides from two mice per group demonstrating baseline histology.
(PDF)

## Author Contributions

**Conceptualization:** Manoj Yadav, Brandon N. D'Souza, Jacquelyn Spathies, Jordan Zeldin, Ian A. Myles.

**Data curation:** Manoj Yadav, Prem Prashant Chaudhary, Grace Ratley, Jacquelyn Spathies, Sundar Ganesan, Jordan Zeldin, Ian A. Myles.

**Formal analysis:** Manoj Yadav, Prem Prashant Chaudhary, Grace Ratley, Sundar Ganesan, Ian A. Myles.

**Funding acquisition:** Ian A. Myles.

**Investigation:** Manoj Yadav, Prem Prashant Chaudhary, Brandon N. D'Souza, Jacquelyn Spathies, Ian A. Myles.

**Methodology:** Manoj Yadav, Prem Prashant Chaudhary, Brandon N. D'Souza, Jacquelyn Spathies, Sundar Ganesan, Jordan Zeldin, Ian A. Myles.

**Project administration:** Ian A. Myles.

**Resources:** Sundar Ganesan, Jordan Zeldin, Ian A. Myles.

**Software:** Prem Prashant Chaudhary, Sundar Ganesan, Jordan Zeldin, Ian A. Myles.

**Supervision:** Ian A. Myles.

**Validation:** Brandon N. D'Souza, Jordan Zeldin, Ian A. Myles.

**Visualization:** Manoj Yadav, Prem Prashant Chaudhary, Grace Ratley, Sundar Ganesan, Ian A. Myles.

**Writing – original draft:** Ian A. Myles.

**Writing – review & editing:** Manoj Yadav, Brandon N. D'Souza, Jacquelyn Spathies, Jordan Zeldin, Ian A. Myles.

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
