## [Decision Letter · Decision Letter 0]

19 Dec 2022

PONE-D-22-31629­­Diisocyanates influence models of atopic dermatitis through direct activation of TRPA1PLOS ONE

Dear Dr. Myles,

Thank you for submitting your manuscript to PLOS ONE. After careful consideration, we feel that it has merit but does not fully meet PLOS ONE’s publication criteria as it currently stands. Therefore, we invite you to submit a revised version of the manuscript that addresses the points raised during the review process.

ACADEMIC EDITOR: The manuscript has been reviewed by one referee. After carefully reading of the whole manuscript and considering the reviewer's comment, we think this story is interesting and therefore invite you to resubmit the manuscript after revision, particularly on the statistics and n numbers for each experiment in the result, which should be detailed in the figure legends. The statistics para in the text should include the detail methods used in the study, not just the software (line 149-line 151). The sample number should be given in figure legends for Figure 3D, Figure 5E-G.  Abbreviations used in figures should also be  indicated in figure legends. In addition, whole manuscript should be carefully proofread.==============================

We look forward to receiving your revised manuscript.

Kind regards,

Sam Xu, PhD, MD, 

Hull York Medical School, UK

Academic Editor

PLOS ONE

Journal Requirements:

"This work was supported by the Intramural Research Program of the National Institute of Allergy and Infectious Diseases (NIAID)."

Additional Editor Comments:

The data is interesting, so we will consider your resubmission after revision. 

Reviewers' comments:

Reviewer's Responses to Questions

**Comments to the Author**

1. Is the manuscript technically sound, and do the data support the conclusions?

Reviewer #1: Partly

2. Has the statistical analysis been performed appropriately and rigorously? 

Reviewer #1: I Don't Know

3. Have the authors made all data underlying the findings in their manuscript fully available?

Reviewer #1: Yes

4. Is the manuscript presented in an intelligible fashion and written in standard English?

Reviewer #1: No

5. Review Comments to the Author

Reviewer #1: The authors have identified isocyanates, and in particular, toluene diisocyanate (TDI), as the pollutant class with the strongest association with AD. They could show isocyanates (e.g., TDI) could disrupt lipid homeostasis, and activate TRPA1 in mice which could directly contribute to AD. The authors presented that TDI-induced skin inflammation in mice and calcium flux in human neurons are dependent on TRPA1 and that TRPA1 blockade can exert therapeutic potential in AD.

The authors are encouraged to take the following points into consideration. The abstract can be formulated better, it is hard to understand from sentences what has been modeled and what has been found.

• The statistic section of the manuscript needs substantial revision. It lacks enough information as to what tests were used to measure or compare which elements.

• Did the author find any sex-related differences in the animal study?

• what is the major limitation of this study?

• What are the translational challenges for the application of TRPA1 inhibitors? any expected or potential side effects in humans?

• Please justify the number of animals used, the use of ARRIVE guidelines, and the application of dosages.

6. PLOS authors have the option to publish the peer review history of their article (what does this mean?). If published, this will include your full peer review and any attached files.

Reviewer #1: No

---

## [Author Response · Author response to Decision Letter 0]

4 Jan 2023

Reviewer #1: The authors have identified isocyanates, and in particular, toluene diisocyanate (TDI), as the pollutant class with the strongest association with AD. They could show isocyanates (e.g., TDI) could disrupt lipid homeostasis, and activate TRPA1 in mice which could directly contribute to AD. The authors presented that TDI-induced skin inflammation in mice and calcium flux in human neurons are dependent on TRPA1 and that TRPA1 blockade can exert therapeutic potential in AD.

The authors are encouraged to take the following points into consideration. The abstract can be formulated better, it is hard to understand from sentences what has been modeled and what has been found.

We thank the reviewer for their comments and insights. We have made the requested changes as detailed below. The abstract has been amended for clarity.

• The statistic section of the manuscript needs substantial revision. It lacks enough information as to what tests were used to measure or compare which elements.

We placed a small amount more detail in the stats section but focused on detailing the calculation used in the figure legends.

• Did the author find any sex-related differences in the animal study?

We verified that the effects of interventions were found in both male and female mice but did not make direct comparisons between them so cannot comment beyond that. We have added such comment to the text.

• what is the major limitation of this study?

We have amended the discussion section that previously detailed out limitations to indicate the major one is that we simply do not know TRPA1 expression differences between age and body site. If (for example) we knew that TRPA1 expression is greater in children and fades with age, that might elucidate why most patients with AD see their skin symptoms resolve as they age.

• What are the translational challenges for the application of TRPA1 inhibitors? any expected or potential side effects in humans?

We have added a comment and a review citation to TRPA1 blockade’s promise and challenges. In brief, there is only a theoretical risk of loss of nociception (which could cause someone to ignore signals to avoid an irritant.

• Please justify the number of animals used, the use of ARRIVE guidelines, and the application of dosages.

We have added this to the methods section under mice.

---

## [Decision Letter · Decision Letter 1]

1 Feb 2023

PONE-D-22-31629R1­­Diisocyanates influence models of atopic dermatitis through direct activation of TRPA1PLOS ONE

Dear Dr. Myles,

Thank you for submitting your manuscript to PLOS ONE. After careful consideration, we feel that it has merit but does not fully meet PLOS ONE’s publication criteria as it currently stands. Therefore, we invite you to submit a revised version of the manuscript that addresses the points raised during the review process.

ACADEMIC EDITOR: The manuscript has been reviewed by two referees. There are still many errors and writing issues. The authors should do careful revision and proofreading. Some paragraphs or figures should be reorganized. ---Here are some examples:

Abstract: The abstract is poorly presented. Too long background information (line 13-line 24). The authors should rewrite it with concise background, methodology and results, and conclusions/implications

Line 69, “10mcL per ear was applied to”.  What unit? How to apply?

Line 81, “t” not “T”, also line 165

Line 82, abbr. “IACUC” should give full name. Check whole manuscript for similar the issue.

Line 87, subscript “2” for CO2

Line 95, leave one space between number and unit

Line 102, Should be “Calcium influx …”

Line 112, change the subheading “bacteriology” and give a proper subheading

Line 119, abbr. “MALDI”?

Line 130, Abbr.: NMDS

Results

There are some errors for citing figure numbers (see reviewer comments as well). The result description should be clear and detailed. The paragraph for each result section should be treated in one figure. The figure 5 seems to be for the text under two subheadings, and the "signalling data" section was listed as half figure in figure 5e-g.

The authors should reorganise the data and figure presentation and give proper text description.

We look forward to receiving your revised manuscript.

Kind regards,

Shang-Zhong Xu, PhD, MD

Academic Editor

PLOS ONE

Reviewers' comments:

Reviewer's Responses to Questions

**Comments to the Author**

1. If the authors have adequately addressed your comments raised in a previous round of review and you feel that this manuscript is now acceptable for publication, you may indicate that here to bypass the “Comments to the Author” section, enter your conflict of interest statement in the “Confidential to Editor” section, and submit your "Accept" recommendation.

Reviewer #1: All comments have been addressed

Reviewer #2: (No Response)

2. Is the manuscript technically sound, and do the data support the conclusions?

Reviewer #1: Yes

Reviewer #2: Partly

3. Has the statistical analysis been performed appropriately and rigorously? 

Reviewer #1: Yes

Reviewer #2: Yes

4. Have the authors made all data underlying the findings in their manuscript fully available?

Reviewer #1: Yes

Reviewer #2: Yes

5. Is the manuscript presented in an intelligible fashion and written in standard English?

Reviewer #1: Yes

Reviewer #2: No

6. Review Comments to the Author

Reviewer #1: The authors have responded to the points raised by the reviewer and accordingly revised the manuscript. Thanks. There is no additional comment.

Reviewer #2: My overall impression is that the authors did not pay much attention to this manuscript. Fig. 2 and 3 are mistakenly cited in the text and data in Fig. 4 and 5 are poorly organized, which all made the manuscript difficult to read and follow.

To improve the scientific quality of this paper, I suggest the authors to provide more results in Fig. 1 as follows, and reorganize the data and description of Fig. 2-5 to make it readable and meaningful.

Fig. 1B-D: Results from WT and TRPA1-/- mice before TDI treatment should be provided. If there is no direct comparison of mouse phenotypes before and after TDI treatment (essentially on the same ears of mice), how could the authors conclude that TDI can induce atopic dermatitis and TRPA1 has a contribution? Even if the TDI-induced model has been established previously, these works should not be neglected as they are important and very easy to do.

Fig. 1E-F: The authors claimed direct activation of TRPA1 by TDI, however the data from Schwann cells are not sufficient to support this conclusion. Patch clamp (ideally) or calcium imaging on TRPA1-overexpreesing and control cells should be performed and a dose-response curve of TDI is essential.

Fig. 1F: What does Pre-Tx mean? It does not seem to be a typo of HC but something else.

7. PLOS authors have the option to publish the peer review history of their article (what does this mean?). If published, this will include your full peer review and any attached files.

Reviewer #1: No

Reviewer #2: No

---

## [Author Response · Author response to Decision Letter 1]

16 Feb 2023

Reviewer replies

Reviewer #2: My overall impression is that the authors did not pay much attention to this manuscript. Fig. 2 and 3 are mistakenly cited in the text and data in Fig. 4 and 5 are poorly organized, which all made the manuscript difficult to read and follow.

To improve the scientific quality of this paper, I suggest the authors to provide more results in Fig. 1 as follows, and reorganize the data and description of Fig. 2-5 to make it readable and meaningful.

We made the error of mislabeling Fig 2 and 3 (they were switched in the upload). The call outs were correct, but we apologize and understand why this flip was confusing. By placing them in proper order we now have the intended flow of 

1 – Establishing TRPA1 mice do not have as severe of AD-like dermatitis after TDI exposure

2 – Blocking TRPA1 improves outcomes in the MC903 model of AD

3 – TRPA1 deletion alters metabolism

4 – TRPA1 blockade alters metabolism

5 – The altered metabolism established in 3 and 4 collate into tyrosine metabolism, which can be further targeted by cell culture model.

Fig. 1B-D: Results from WT and TRPA1-/- mice before TDI treatment should be provided. If there is no direct comparison of mouse phenotypes before and after TDI treatment (essentially on the same ears of mice), how could the authors conclude that TDI can induce atopic dermatitis and TRPA1 has a contribution? Even if the TDI-induced model has been established previously, these works should not be neglected as they are important and very easy to do.

We now present data from WT and TRPA1-/- mice without TDI exposure. As has been established in the prior use of these animals – there are no obvious baseline differences. We understand the request for baseline difference, but seeing none we continue to conclude that this established model is correct and that the TDI exposure’s failure to induce a similar degree of inflammation in the knockout mice supports our conclusion that TDI-induced dermatitis is partially dependent on TRPA1.

Fig. 1E-F: The authors claimed direct activation of TRPA1 by TDI, however the data from Schwann cells are not sufficient to support this conclusion. Patch clamp (ideally) or calcium imaging on TRPA1-overexpreesing and control cells should be performed and a dose-response curve of TDI is essential.

We now present three different doses of TDI to demonstrate the dose response curve requested and thank the reviewer for this recommendation. However, we continue to present data that the calcium influx is dependent on TRPA1. Overexpressing TRPA1 would not impact these conclusions. If overexpression of TRPA1 increased flux, indeed that would support the notion that it was involved – but failure to do so would not comment on TRPA1s role since the rate limiting step of calcium influx may not be the level of receptor expression. Our findings are clear – adding TDI in the presence of a specific TRPA1-blocker negates calcium influx. Given that we now also show that this occurs in a dose dependent manner, we do not feel that the patch clamp or overexpression would change the conclusions presented (save a claim that Ca influx would not carry an obvious change in conductivity).

Fig. 1F: What does Pre-Tx mean? It does not seem to be a typo of HC but something else.

Tx is an established abbreviation for “treatment” for physicians, but this was left in error. We apologize and have fixed the figure, so it is Pre-HC to match the remainder of the figure.

Editor replies:

Abstract: The abstract is poorly presented. Too long background information (line 13-line 24). The authors should rewrite it with concise background, methodology and results, and conclusions/implications

The background portion of the abstract has been amended.

Line 69, “10mcL per ear was applied to”. What unit? How to apply?

We are not sure what is being asked by “what unit” since mcL is a well established unit of volume, but we have spelled out microliter and added “topically” to the description of how this was applied “to the ear”.

Line 81, “t” not “T”, also line 165

This has been changed

Line 82, abbr. “IACUC” should give full name. Check whole manuscript for similar the issue.

This has been added

Line 87, subscript “2” for CO2

This has been changed

Line 95, leave one space between number and unit

This has been changed

Line 102, Should be “Calcium influx …”

This has been changed although technically calcium outflux would also be visible, thus “flux” seemed the most appropriate term.

Line 112, change the subheading “bacteriology” and give a proper subheading

We changed this to “bacterial culture”

Line 119, abbr. “MALDI”?

This has been added

Line 130, Abbr.: NMDS

This has been added

Results

There are some errors for citing figure numbers (see reviewer comments as well). The result description should be clear and detailed. The paragraph for each result section should be treated in one figure. The figure 5 seems to be for the text under two subheadings, and the "signalling data" section was listed as half figure in figure 5e-g.

We made the error of mislabeling Fig 2 and 3 (they were switched in the upload). The call outs are correct, but we apologize and understand why this flip was confusing. We have moved the section on figure 5 to one paragraph. We have left Fig 3 and 4 combined because they are highly related and are only separate because combining them would render the text too small to read.

---

## [Editor Report · Decision Letter 2]

21 Feb 2023

­­Diisocyanates influence models of atopic dermatitis through direct activation of TRPA1

PONE-D-22-31629R2

Dear Dr. Myles,

We’re pleased to inform you that your manuscript has been judged scientifically suitable for publication and will be formally accepted for publication once it meets all outstanding technical requirements.

Kind regards,

Shang-Zhong Xu, PhD, MD

Academic Editor

PLOS ONE
---

## [Editor Report · Acceptance letter]

23 Feb 2023

PONE-D-22-31629R2 

­­Diisocyanates influence models of atopic dermatitis through direct activation of TRPA1 

Dear Dr. Myles:

I'm pleased to inform you that your manuscript has been deemed suitable for publication in PLOS ONE. Congratulations! Your manuscript is now with our production department. 

Kind regards, 

on behalf of

Dr. Shang-Zhong Xu 

Academic Editor

PLOS ONE